# Benthic primary production in an upwelling-influenced coral reef, Colombian Caribbean

Corvin Eidens[1,2], Elisa Bayraktarov[2,5], Torsten Hauffe[1], Valeria Pizarro[3], Thomas Wilke[1] and Christian Wild[2,4]

[1] Department of Animal Ecology & Systematics, Justus Liebig University Giessen, Germany
[2] Coral Reef Ecology Group (CORE), Leibniz Center for Tropical Marine Ecology, Bremen, Germany
[3] Center of Excellence in Marine Sciences (CEMarin), Santa Marta, Colombia
[4] Faculty of Biology and Chemistry, University of Bremen, Germany
[5] Current affiliation: Global Change Institute, The University of Queensland, Brisbane, Australia

Corresponding author
Corvin Eidens,
corvin.eidens@allzool.bio.uni-giessen.de

## ABSTRACT

In Tayrona National Natural Park (Colombian Caribbean), abiotic factors such as light intensity, water temperature, and nutrient availability are subjected to high temporal variability due to seasonal coastal upwelling. These factors are the major drivers controlling coral reef primary production as one of the key ecosystem services. This offers the opportunity to assess the effects of abiotic factors on reef productivity. We therefore quantified primary net ($P_n$) and gross production ($P_g$) of the dominant local primary producers (scleractinian corals, macroalgae, algal turfs, crustose coralline algae, and microphytobenthos) at a water current/wave-exposed and -sheltered site in an exemplary bay of Tayrona National Natural Park. A series of short-term incubations was conducted to quantify $O_2$ fluxes of the different primary producers during non-upwelling and the upwelling event 2011/2012, and generalized linear models were used to analyze group-specific $O_2$ production, their contribution to benthic $O_2$ fluxes, and total daily benthic $O_2$ production. At the organism level, scleractinian corals showed highest $P_n$ and $P_g$ rates during non-upwelling (16 and 19 mmol $O_2$ m$^{-2}$ specimen area h$^{-1}$), and corals and algal turfs dominated the primary production during upwelling (12 and 19 mmol $O_2$ m$^{-2}$ specimen area h$^{-1}$, respectively). At the ecosystem level, corals contributed most to total $P_n$ and $P_g$ during non-upwelling, while during upwelling, corals contributed most to $P_n$ and $P_g$ only at the exposed site and macroalgae at the sheltered site, respectively. Despite the significant spatial and temporal differences in individual productivity of the investigated groups and their different contribution to reef productivity, differences for daily ecosystem productivity were only present for $P_g$ at exposed with higher $O_2$ fluxes during non-upwelling compared to upwelling. Our findings therefore indicate that total benthic primary productivity of local autotrophic reef communities is relatively stable despite the pronounced fluctuations of environmental key parameters. This may result in higher resilience against anthropogenic disturbances and climate change and Tayrona National Natural Park should therefore be considered as a conservation priority area.

## INTRODUCTION

The majority of ecosystems depend on primary production. Photoautotrophs convert light energy into chemical energy by photosynthesis, creating the energetic base of most food webs in terrestrial as well as aquatic environments (*Chapin et al., 2011*; *Valiela, 1995*). Among other coastal ecosystems such as mangrove forests, seagrass beds, salt marshes, and kelp forests, coral reefs belong to the most productive ecosystems in the world (*Gattuso, Frankignoulle & Wollast, 1998*; *Hatcher, 1988*). Productivity investigation on coral reefs started in the mid-20th century (*Odum & Odum, 1955*; *Sargent & Austin, 1949*) and nowadays, coral reefs are among the best understood marine benthic communities in terms of primary production (*Gattuso, Frankignoulle & Wollast, 1998*; *Hatcher, 1988*; *Hatcher, 1990*; *Kinsey, 1985*).

It was long assumed that coral reef productivity is relatively balanced as tropical coral reefs typically thrive under relatively stable abiotic conditions (*Hubbard, 1996*; *Kleypas, McManus & Menez, 1999*; *Sheppard, Davy & Pilling, 2009*), including light (*Achituv & Dubinsky, 1990*; *Darwin, 1842*; *Falkowski, Jokiel & Kinzie, 1990*), water temperature (*Coles & Fadlallah, 1991*; *Dana, 1843*; *Veron, 1995*), salinity (*Andrews & Pickard, 1990*; *Coles & Jokiel, 1992*), and inorganic nutrient availability (*D'Elia & Wiebe, 1990*; *Szmant, 1997*).

Nevertheless, coral reefs also occur in seasonal upwelling-affected regions such as the Arabian Sea off Oman (*Glynn, 1993*), the Eastern Tropical Pacific off Panamá and Costa Rica (*Cortés & Jiménez, 2003*; *Glynn & Stewart, 1973*), and the Colombian Caribbean (*Geyer, 1969*). Whereas several studies focused on the seasonality of benthic primary production in coral reefs at different latitudes (*Adey & Steneck, 1985*; *Falter et al., 2012*; *Kinsey, 1985*), variability in primary production of seasonal upwelling-affected coral reefs remains largely unknown.

The Tayrona National Natural Park (TNNP) at the Caribbean coast of Colombia is highly influenced by the Southern Caribbean upwelling system (*Andrade & Barton, 2005*; *Rueda-Roa & Muller-Karger, 2013*), causing seasonal fluctuations in water temperature, salinity, and inorganic nutrient concentrations, among others (Table 1, see also *Bayraktarov, Pizarro & Wild, 2014*). Here, the abundance and community composition of benthic algae were shown to exhibit upwelling-related seasonality (*Diaz-Pulido & Garzón-Ferreira, 2002*; *Eidens et al., 2012*). The area thereby provides an excellent opportunity to investigate the effects of seasonal coastal upwelling events on the key ecosystem service productivity in coral reefs under changing *in situ* conditions. The results of a preliminary study conducted by *Eidens et al. (2012)* indicated that benthic primary production in TNNP differed between the upwelling in 2010/2011 and the consecutive non-upwelling season, suggesting a generally positive effect of upwelling conditions on major benthic autotrophs in the area. However, after unusually strong El Niño-Southern Oscillation (ENSO) events in 2010, the area experienced a moderate coral bleaching before the upwelling in 2010/2011 (*Bayraktarov et al., 2013*; *Hoyos et al., 2013*), and productivity measurements during upwelling in 2010/2011 may not be representative. To test for patterns in benthic primary production during a typical seasonal cycle, we here quantified benthic primary production before and at the end of the upwelling event in 2011/2012 (hereafter referred

**Table 1  Seasonality in water temperature, salinity and nitrate availability in Gayraca Bay.** Mean values (±SD) at the exposed and sheltered sites and a water depth of 10 m for upwelling (December–April) and non-upwelling (May–November) periods from 2010–2013.

| Variables | Non-upwelling | | Upwelling | | Range |
|---|---|---|---|---|---|
| | Exposed | Sheltered | Exposed | Sheltered | |
| Temperature (°C) | 28.5 ± 1.7 | 28.7 ± 1.7 | 25.7 ± 2.6 | 25.7 ± 2.5 | 20.5–30.0 |
| Salinity | 35.3 ± 1.5 | 35.3 ± 1.2 | 37.1 ± 1.1 | 37.0 ± 0.8 | 32.6–38.5 |
| Nitrate ($\mu$mol L$^{-1}$) | 0.26 ± 0.20 | 0.32 ± 0.16 | 1.31 ± 0.95 | 1.34 ± 0.99 | nd–3.59 |

**Notes.**
nd, below detection level.

to as non-upwelling and upwelling, respectively). To allow for comparisons of productivity between investigated groups, we further estimated surface area-specific productivity rates as suggested by *Naumann et al. (2013)* and analyzed the data using generalized linear models.

Therefore, the goals of the study were to (1) identify dominant functional groups of benthic primary producers and their relative benthic cover at a current/wave-exposed (EXP) and -sheltered (SHE) site in one exemplary bay of TNNP, (2) quantify $O_2$ fluxes of all dominant benthic primary producers and apply 3D surface area estimations, and hence (3) estimate the specific contribution of each group to total benthic $O_2$ fluxes.

# MATERIALS AND METHODS

## Study site and sampling seasons

This study was conducted in Gayraca Bay (11.33°N, 74.11°W), one of several smaller bays in TNNP, located near the city of Santa Marta (Fig. 1). The continental shelf in the area is relatively narrow due to the proximity to the Sierra Nevada de Santa Marta–the world's highest coastal mountain range. The TNNP contains small fringing coral reefs reaching to a water depth of ∼30 m (*Garzón-Ferreira, 1998*; *Garzón-Ferreira & Cano, 1991*). The region is subjected to strong seasonality caused by the Caribbean Low-Level Jet of northeast (NE) trade winds (*Andrade & Barton, 2005*; *Salzwedel & Müller, 1983*), resulting in two major seasons; a dry season from December to April and a rainy season from May to November (*Garzón-Ferreira, 1998*; *Salzwedel & Müller, 1983*). Whereas the rainy season (non-upwelling) is characterized by low wind velocities (mean 1.5 m s$^{-1}$) (*Garzón-Ferreira, 1998*) and high precipitation (>80% of the annual rainfall) (*Salzwedel & Müller, 1983*), during the dry season (upwelling), strong winds prevail (mean 3.5 m s$^{-1}$, max 30 m s$^{-1}$) (*Herrmann, 1970*; *Salzwedel & Müller, 1983*) resulting in a seasonal coastal upwelling. The upwelling-related changes in key water parameters are well characterized by the comprehensive study of *Bayraktarov, Pizarro & Wild (2014)*. During upwelling, water temperature can decrease to 20 °C while salinity and nitrate availability increase up to 39 and 3.59 $\mu$mol L$^{-1}$, respectively (Table 1). Water currents triggered by prevailing winds predominantly move from NE to SW, and a clear gradient in wave exposure between the exposed western (EXP) and -sheltered northeastern (SHE) sides of the bay can be observed (*Bayraktarov, Bastidas-Salamanca & Wild, 2014*; *Werding & Sánchez, 1989*). The study

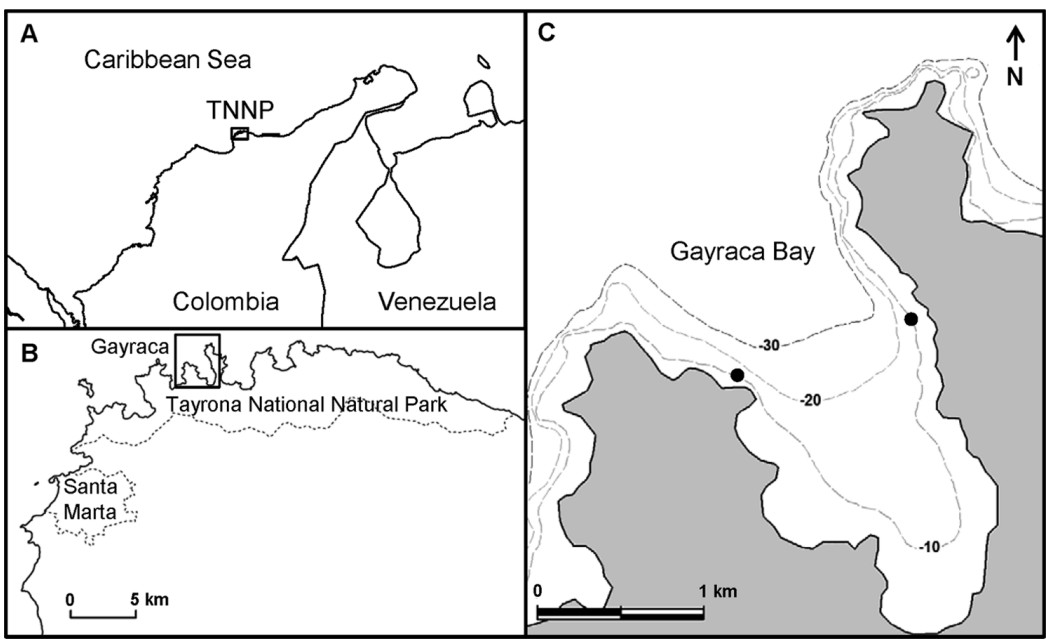

**Figure 1 Location of study sites.** (A) Location of Tayrona National Natural Park (TNNP) at the Caribbean coast of Colombia. (B) Location of Gayraca Bay within TNNP (dashed lines–national park border and expansion of the city of Santa Marta. (C) Gayraca Bay. The investigation sites at the current-exposed western part and the sheltered north-eastern part are indicated by black dots (dashed lines–isobaths). Source of map: *INVEMAR (2012)*.

was carried out during non-upwelling in 2011 (1st November–2nd December 2011) and during the consecutive upwelling event (20th March–29th March 2012), allowing for the investigation of the influence of seasonality on benthic primary production.

## Benthic assessment

For the assessment of benthic community structure, the dominant groups of benthic primary producers and the percentage of benthic cover were identified at EXP and SHE prior to primary production measurements using line point intercept transects at a water depth of 10 m (50 m length, $n = 3$), modified from *Hodgson et al. (2004)*. Benthic cover was monitored at 0.5 m intervals directly below the measurement points (101 data points per transect). The dominant benthic autotrophs at the study sites consisted of scleractinian corals, frondose macroalgae, algal turfs (multispecific assemblage of primarily filamentous algae of up to 1 cm height, *sensu Steneck (1988)*), crustose coralline algae (CCA), and sand potentially associated with microphytobenthos. These categories represented $97 \pm 1\%$ of the total seafloor coverage at SHE and $91 \pm 2\%$ at EXP and were therefore selected as representative primary producers for the subsequent incubation experiments. During benthic community assessment, rugosity was determined at both sites using the chain method described by *Risk (1972)*. Rugosity was quantified along three 10 m sub-transects within each of the 50 m transects and were used to calculate the rugosity factor for each study site as described by *McCormick (1994)* (SHE: $1.53 \pm 0.12$, EXP: $1.32 \pm 0.13$).

**Table 2 Water temperature and light intensity during incubation experiments at sampling sites and in incubation containers.** All values are in mean ± SD. Data in parentheses represent water temperature and light intensity and at the end of the upwelling event in 2010/2011.

| | Non-upwelling | | Upwelling | |
|---|---|---|---|---|
| | *In situ* | Incubations | *In situ* | Incubations |
| Temperature (°C) | 29.1 ± 0.2 | 28.6 ± 0.5 | 25.3 ± 0.3 (26.1 ± 0.2) | 25.4 ± 0.6 (26.5 ± 0.4) |
| Light intensity (PAR µmol photons m$^{-2}$ s$^{-1}$) | 146 ± 47 | 154 ± 40 | 230 ± 58 (234 ± 78) | 257 ± 69 (248 ± 71) |

## Sampling of organisms

Specimens of scleractinian corals, macroalgae, algal turfs, and CCA as well as sand samples, from $10 \pm 1$ m water depth were used for quantification of $O_2$ fluxes (see Table S1 for number of replicates). All samples were brought to the water surface in Ziploc® bags and transported directly to the field laboratory. Scleractinian corals of the genera *Montastraea* (including the species *M. faveolata, M. franksi* and *M. annularis*, currently belonging to the genus *Orbicella*; *Budd et al., 2012*) and *Diploria* (including *D. strigosa*, currently belonging to the genus *Pseudodiploria Budd et al., 2012*) accounted for more than 80% of the total coral cover at the study sites and were therefore used as representative corals in our study. Coral specimens were obtained from the reef using hammer and chisel, fragmented with a multifunction rotary tool (8,200–2/45; mean fragment surface area: $13.16 \pm 7.96$ cm$^2$, Dremel Corp.), and fixed on ceramic tiles using epoxy glue (Giesemann GmbH, Aquascape). After fragmentation, specimens were returned to their natural habitat and left to heal for one week prior to the incubation experiments. Algae of the genus *Dictyota* (mainly *D. bartayresiana*) amounted to nearly 100% of macroalgal cover. Therefore small bushes of *Dictyota* spp. (surface area $1.86 \pm 0.88$ cm$^2$) were used as representatives for macroalgae. Macroalgae were transferred to a storage tank (volume: 500 L in which water was exchanged manually 3–5 times per day and water temperature was within the ranges of incubation experiments; Table 2) one day before incubation experiments and left to heal. All other functional groups were incubated immediately after sampling. Rubble overgrown by algal turfs and CCA served as samples for the respective functional group (surface area covered by the organisms: $15.63 \pm 10.80$ cm$^2$ and $7.48 \pm 3.60$ cm$^2$, respectively). For sand samples, custom-made mini corers with defined surface area (1.20 cm$^2$) and sediment core depth (1.0 cm) were used. All necessary permits (DGI-SCI-BEM-00488) were obtained by Instituto de Investigaciones Marinas y Costeras (INVEMAR) in Santa Marta, Colombia which complied with all relevant regulations.

## Surface area quantification

Digital photographs of coral specimens were used to quantify planar projected surface areas of samples by image-processing software (ImageJ, V. 1.46r, National Institute of Health). The 3D surface area of the samples was estimated via multiplication of the planar projected surface areas by the genera-specific 2D to 3D surface area conversion factors derived from computer tomography measurements of *Diploria* and *Montastraea* skeletons ($2.28 \pm 0.16$ and $1.34 \pm 0.56$, respectively), as described by *Naumann et al. (2009)*. Planar

leaf area of spread out macroalgal specimens was likewise quantified by digital image analysis and multiplied by the factor 2 to obtain 3D surface area of the samples. Image analysis of *in situ* photographs and whole spread out macroalgal thalli were used to obtain covered substrate areas (2D surface) as well as 3D surface areas in order to calculate the 2D to 3D conversion factor for macroalgae ($4.29 \pm 0.82$). This conversion factor was used to correct for the overlap of macroalgal tissue. The 2D surface area of algal turf samples was determined by image analysis of digital photographs. For CCA, the simple geometry method described by *Naumann et al. (2009)* was used to estimate the surface area of overgrown pieces of rubble. The obtained surface areas were related to the planar projected surface area of the samples to generate 2D to 3D conversion factors for CCA ($2.10 \pm 0.89$). Specimen surface area for sand samples was defined by the size of the utilized mini corer ($1.20\ \text{cm}^2$).

## INCUBATION EXPERIMENTS

Prior to incubation experiments, water temperature (°C) and light intensity (lx) were monitored at the sampling sites with intervals of 2 min using light and temperature loggers (Onset HOBO Pendant UA-002-64) in order to adjust light and temperature during incubations to *in situ* conditions. The availability of light during light incubations was adjusted to the *in situ* light regimes using net cloth (Table 2). Temperature and light intensity was continuously monitored during incubations as described above. Light intensities were converted to photosynthetically active radiation (PAR, $\mu$mol photons m$^{-2}$ s$^{-1}$, 400–700 nm) using the approximation of *Valiela (1995)*. Light availability was generally higher during the upwelling event ($t$-test, $p < 0.001$; Table 2), whereas water temperatures were higher during non-upwelling ($t$-test, $p < 0.001$; Table 2). Quantification of photosynthetic activity for macroalgae, CCA, and microphytobenthos were performed in air-tight glass containers with volumes of 60 mL, whereas for corals and algal turfs, containers with volumes of 600 mL were utilized. For all incubations, we used freshly collected seawater from Gayraca Bay. To ensure independence between the samples, each specimen was incubated in a distinct container. The containers were placed in cooling boxes filled with seawater to maintain constant *in situ* water temperature (Table 2). For dark incubations during daytime, the above mentioned methodology was used, but cooling boxes were closed with opaque lids to prevent light penetration. Comparability among measurements was assured by carrying out all light incubations on cloudless days between 10 am and 2 pm. For each group of primary producers, one light and one dark incubation were performed within each study period. Incubation containers containing only seawater served as blank controls to quantify photosynthetic activity and respiration of microbes in the water column. Physiological damage of the investigated specimens by hypoxic or hyperoxic conditions were prevented by keeping the incubation times as short as possible (light incubations: 30–60 min and dark incubations: 120 min as suggested by *Jantzen et al., 2008*; *Mass et al., 2010b*; *Jantzen et al., 2013*). Dissolved O$_2$ concentrations in the incubation water within the glass containers were quantified before incubations and after removing the specimens at the end of each incubation using an optode

(Hach Lange, HQ 40). Before $O_2$ measurements, the incubation medium was gently stirred with the optode sensor allowing a homogenization of the water column. Experiments were conducted in closed, non-mixed incubation chambers in order to avoid additional contamination sources and to provide the most conservative estimates of $O_2$ production rates of benthic primary producers as suggested by *Haas et al. (2011)* and *Naumann et al. (2013)*. This also ensured higher measurement accuracy, as water movement during incubations may affect gas transfer velocities across the surface boundary of the incubation chambers (*Wu, Barazanji & Johnson, 1997*) and allowed us to compare our results with previous incubation studies (e.g., *Haas et al., 2011*; *Jantzen et al., 2013*; *Naumann et al., 2013*). Nevertheless, since it is well known that water flow enhances $O_2$ fluxes and thereby photosynthesis (*Mass et al., 2010a*), the results of the field incubations should be regarded as conservative estimates of *in situ* $O_2$ fluxes and interpreted accordingly.

## Data analyses and statistics

To quantify net $O_2$ production ($P_n$) and respiration of functional groups, $O_2$ concentration before incubations was subtracted from concentration after incubations and blank control values were subtracted from the measured $O_2$ fluxes. Individual gross $O_2$ production ($P_g$) of investigated functional groups was calculated by adding values of $P_n$ and respiration; individual $O_2$ fluxes were expressed as mmol $O_2$ m$^{-2}$ specimen surface area h$^{-1}$.

The contribution of each functional group to total reef production (given as: mmol $O_2$ m$^{-2}$ seafloor area h$^{-1}$) was estimated as follows:

$$c_i = p_i \, s_i \, b_i \, r$$

taking into account the individual production rates ($p_i$), the respective mean 2D to 3D surface conversion factor ($s_i$), group-specific benthic coverage ($b_i$) as well as the rugosity factor ($r$). Estimation of total daily benthic productivity was furthermore calculated by summing up the contribution of the investigated groups and extrapolating the incubation periods to a 12 h light and 12 h dark cycle.

After testing for normal distribution (Kolmogorov-Smirnoff test) and homogeneity of variances (Levene test), benthic coverage of functional groups were analyzed using two-way ANOVA and Bonferroni's *post hoc* tests to detect possible effects of season (upwelling vs. non-upwelling) and site (EXP vs. SHE) and their interaction on benthic cover.

We tested the influence of benthic groups, season, wave exposure, and their interactions on $O_2$ productivity by generalized linear models (GLMs) for individual $P_n$ and $P_g$ of the investigated groups, their contribution to reef metabolism as well as total benthic productivity. We used Markov-chain Monte Carlo (MCMC) estimations of GLM regression coefficients. In traditional Frequentist statistics, the parameters of interest (i.e., the $O_2$ productivity describing regression coefficients) are estimated just once (e.g., using Maximum-Likelihood) and their significance is inferred indirectly based on a test-statistic. In contrast, Bayesian methods reallocate the coefficients across a set of possible candidates during each MCMC generation (*Kruschke, 2011*). If the bulk of these

values, that is the 95% highest posterior density (HPD), does not include zero, one can directly conclude that the regression coefficient is credible different than zero and an effect on $O_2$ productivity exists. Moreover, we here performed pair-wise comparisons between benthic groups at different sites and seasons, traditionally being performed by *post-hoc* testing with $P$-value correction for preventing false positive results. A Bayesian GLM does not suffer this drawback because difference of groups can be directly estimated by the posterior (*Kruschke, 2011*). Again, there is credible evidence in non-equal group-means, if the posterior-based 95% HPD interval of the group differences does not include zero. Model performance for all 19 possible combinations of the three independent variables and their interactions was assessed by the deviance information criterion (DIC), a Bayesian measure of model fit that penalizes complexity (*Spiegelhalter et al., 2002*). In this information theory based model selection, often there is not a single best model describing the data. Therefore, averaging of regression coefficients for all models within $\Delta DIC < 2$ of the best one (*Johnson & Omland, 2004*) was performed according to DIC weights (i.e., support for the respective regression model).

Here, Bayesian GLMs using the MCMCglmm package (*Hadfield, 2010*) for the R 3.0.3 environment for statistical computing (*R Core Team, 2014*) with a Gaussian error distribution were applied. Prior to the analyses, the mean–variance relationship of measured $O_2$ flux was stabilized by power transformation (*Yeo & Johnson, 2000*). Visual inspection of preliminary GLMs with default weakly informative priors showed high autocorrelation in their posterior distribution. Thus to infer the posterior distribution of the final analyses, we ignored the first 50,000 estimates as burn-in and sampled every 5th out of 650,000 MCMC generations.

All values are represented as mean $\pm$ standard deviation (SD) if not noted otherwise.

## RESULTS

### Benthic community composition

At EXP, scleractinian corals dominated the benthic community during non-upwelling and upwelling ($41 \pm 12$ and $39 \pm 12\%$, respectively; Fig. 2). At SHE, corals, algal turf, and sand cover was similar during non-upwelling ($24 \pm 3\%$, $26 \pm 6\%$, and $25 \pm 13\%$, respectively), while during upwelling, macroalgae exhibited highest benthic cover ($47 \pm 3\%$, Fig. 2). During the entire study period, coral and CCA cover was significantly higher at EXP than at SHE, whereas sand showed a contrary pattern with significantly more coverage at SHE (Fig. 2). Macroalgae was the only group where interaction between sites and seasons occurred with significantly higher cover at SHE and higher abundances during upwelling at both sites (Fig. 2). CCA cover also differed between the seasons, showing a significant decrease during the upwelling event (Fig. 2).

### $O_2$ fluxes of organisms

More complex Bayesian GLMs, including interactions among the three independent variables season, benthic group, and site, described individual $O_2$ fluxes better than simple models (For details see Table S2).

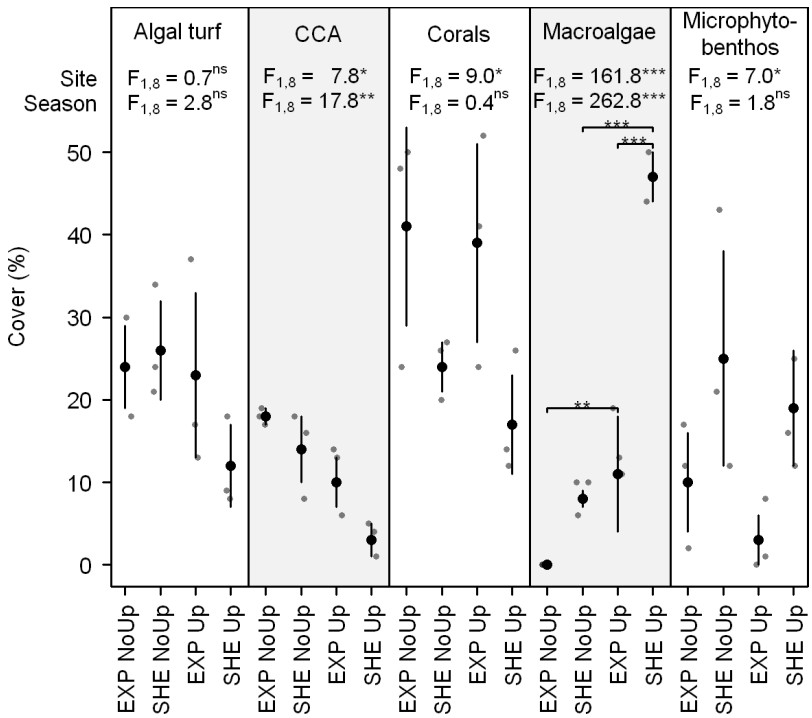

**Figure 2 Benthic cover of dominant benthic groups.** Jitter plot of grey dots indicates benthic coverage of functional groups quantified by line transects. Superimposed black points and error bars represent the mean ± standard deviation. *F*-values refer to the results of two-way analyses of variance with Site and Season as main effects. If interactions between main effects were significant, pair-wise *post hoc* tests were applied. Significance levels are *$p < 0.05$, **$p < 0.01$, ***$p < 0.001$. Abbreviations: EXP, exposed; SHE, sheltered; NoUp, non-upwelling; Up, upwelling.

Of all investigated functional groups, scleractinian corals had highest individual net ($P_n$) and gross production ($P_g$), followed by algal turfs, macroalgae, CCA, and microphytobenthos (Fig. 3; see also Table S3 for detailed results of all pair-wise comparisons). Regarding spatial differences in individual productivity, significant differences were detected for algal turfs and CCA. During upwelling, $P_n$ of algal turfs and $P_g$ of CCA was higher at SHE than EXP. On the contrary, during non-upwelling, $P_n$ and $P_g$ of CCA was higher at EXP (Fig. 3).

Temporal differences in $O_2$ production were detected for corals, algal turfs, and CCA (Fig. 3). Whereas $P_n$ of scleractinian corals on both sites and $P_g$ of CCA at EXP were higher during non-upwelling, $P_g$ of CCA at SHE as well as $P_n$ and $P_g$ of algal turfs at both sites showed an opposite pattern with higher productivity rates during upwelling (Fig. 3).

### Contribution of organism-induced $O_2$ fluxes to total reef $O_2$ production

As in the case of individual $O_2$ fluxes, contribution and total reef production were better explained by GLMs of higher complexity (Table S2).

Contribution of functional groups to benthic productivity exhibited similar pattern than individual productivity with corals contributing generally most to total reef $P_n$

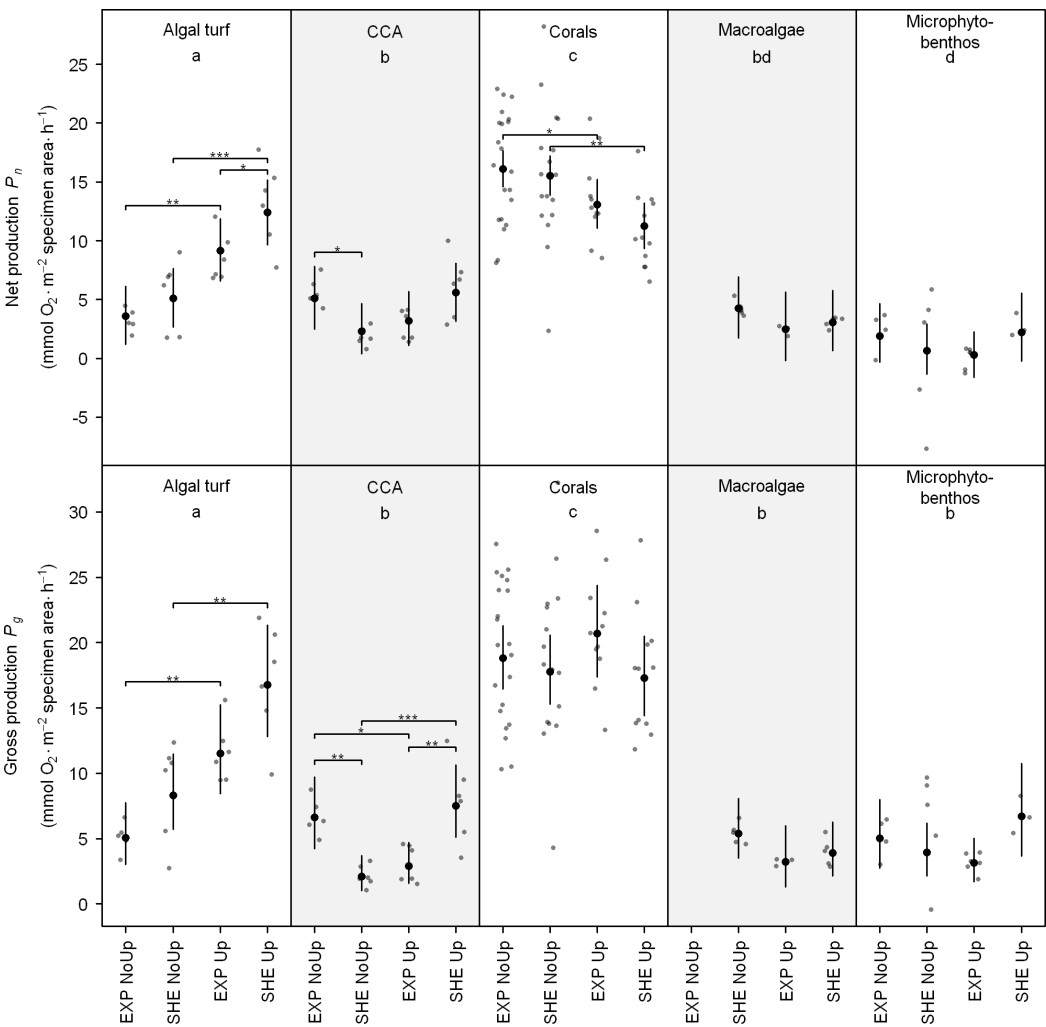

**Figure 3 Individual net and gross production of benthic functional groups.** Jitter plot of grey dots indicates measured $O_2$ fluxes. Superimposed black points and error bars represent the mean and 95% confidence interval according to the Bayesian generalized linear model. Equal lowercase letters indicate no differences in mean productivity among benthic groups and brackets display differences within groups. Significance levels are *$p$MCMC < 0.05, **$p$MCMC < 0.01, ***$p$MCMC < 0.001. Abbreviations: EXP, exposed; SHE, sheltered; NoUp, non-upwelling; Up, upwelling.

and $P_g$, but macroalgae contributed most to benthic $P_n$ and $P_g$ at SHE at the end of upwelling (Fig. 4; see also Table S3 for detailed results of all pair-wise comparisons).

Significant spatial differences in contribution to total benthic $P_n$ within functional groups were detected for corals, algal turf, and macroalgae, and spatial differences for $P_g$ were present in all investigated groups except CCA (Fig. 4). At EXP, Corals contributed more to total $P_n$ and $P_g$ during non-upwelling and upwelling (Fig. 4). At SHE, contributions of macroalgae ($P_n$ and $P_g$) and microphytobenthos ($P_g$) were higher only during upwelling, and algal turfs contributed more to $P_g$ at SHE during non-upwelling (Fig. 4).

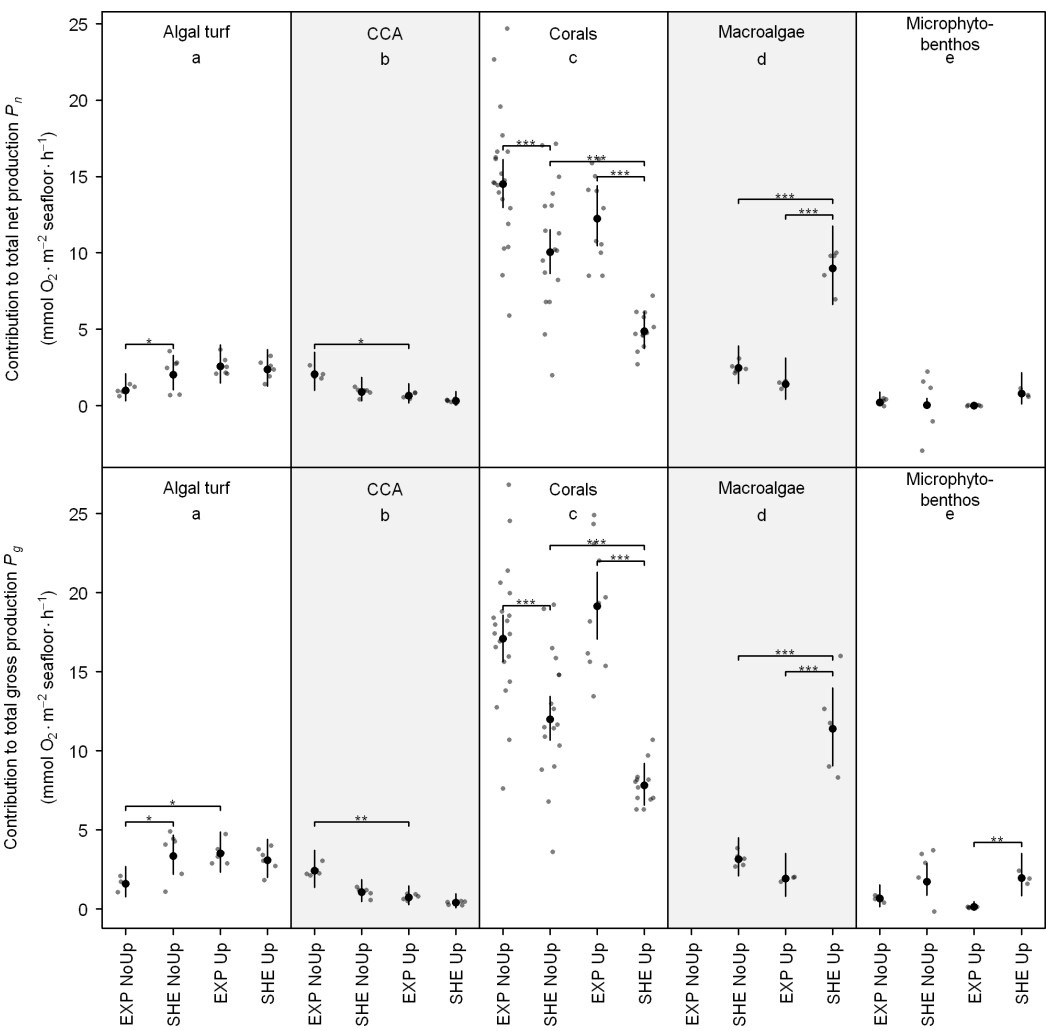

**Figure 4 Contribution of functional groups to benthic net and gross production.** Jitter plot of grey dots indicates measured $O_2$ fluxes. Superimposed black points and error bars represent the mean and 95% confidence interval according to the Bayesian generalized linear model. Equal lowercase letters indicate no differences in mean productivity among benthic groups and brackets display differences within groups. Significance levels are $*p$MCMC $< 0.05$, $**p$MCMC $< 0.01$, $***p$MCMC $< 0.001$. Abbreviations: EXP, exposed; SHE, sheltered; NoUp, non-upwelling; Up, upwelling.

Temporal differences in contribution to total benthic productivity within the investigated groups were present for corals, macroalgae, CCA (for $P_n$ and $P_g$), and for algal turfs (only $P_g$) (Fig. 5). During non-upwelling, corals contributed more to the total productivity at SHE and CCA at EXP, whereas during upwelling, macroalgae contributed more to the total productivity at SHE and algal turf at EXP (Fig. 4).

Regarding the total daily benthic $O_2$ fluxes (Fig. 4), no spatial differences between EXP and SHE were detected, neither during non-upwelling nor during upwelling (see also Table S3 for detailed results of all pair-wise comparisons). During the study, significant temporal differences were only present for $P_g$ at the exposed site with higher $O_2$ fluxes during the upwelling in 2011/2012 compared to non-upwelling (Fig. 5). Comparing

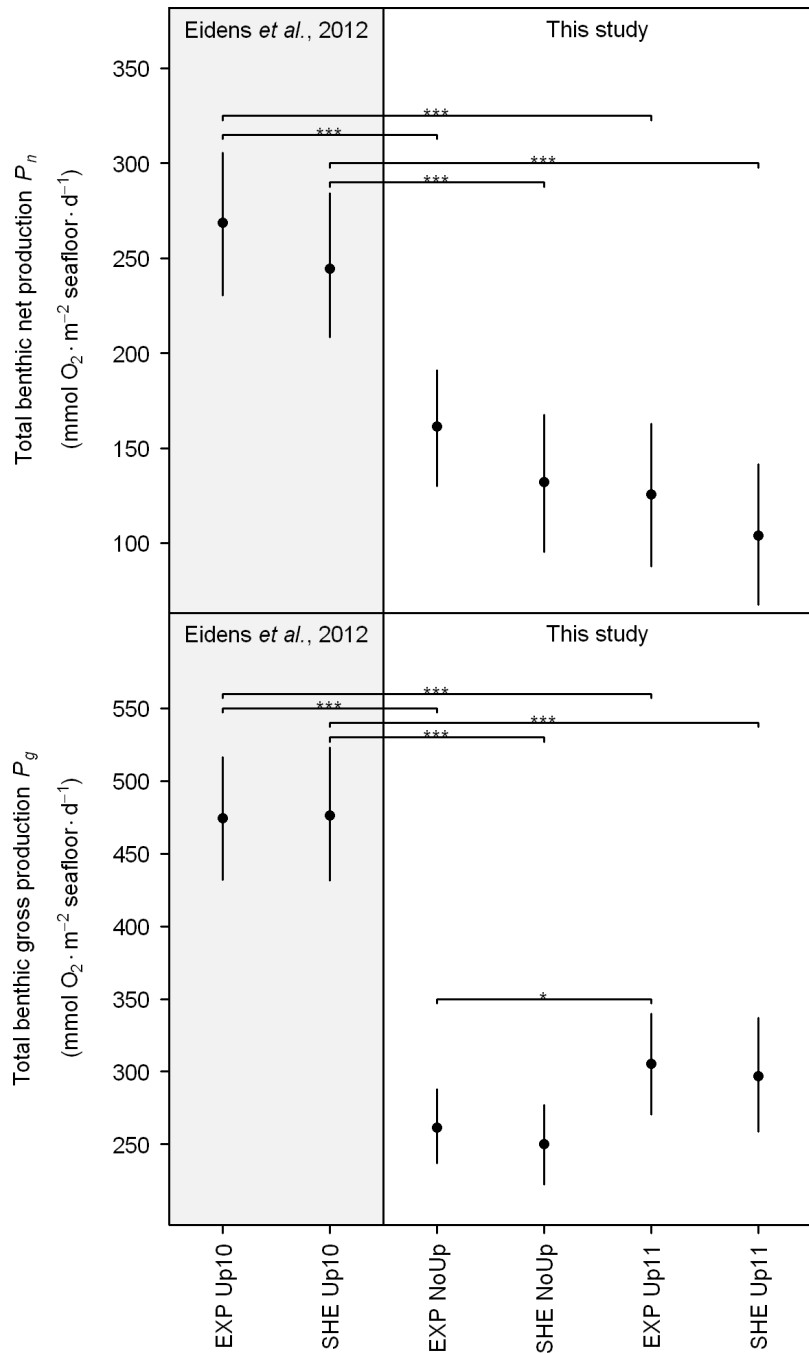

**Figure 5 Total benthic net and gross production.** Black points and error bars represent the mean and 95% confidence interval according to the Bayesian generalized linear model. Brackets display differences between seasons. Significance levels are $*p\text{MCMC} < 0.05$, $**p\text{MCMC} < 0.01$, $***p\text{MCMC} < 0.001$. Abbreviations: EXP, exposed; SHE, sheltered; NoUp, non-upwelling; Up10, upwelling 2010/2011; Up11, upwelling 2011/2012.

total benthic productivity during the upwelling event in 2010/2011 with the subsequent non-upwelling and upwelling, $P_n$ and $P_g$ were significantly higher during the upwelling 2010/2011 for all comparisons (Fig. 5).

## DISCUSSION

### O$_2$ fluxes of organisms

Individual mean $P_n$ and $P_g$ were generally highest for corals at both sites during the study periods ($P_n$: 11.2–16.1 and $P_g$: 17.4–20.8 mmol O$_2$ m$^{-2}$ specimen area h$^{-1}$). These high productivity rates of corals compared to other investigated primary producers (see Fig. 3) may be attributed to the mutualistic relationship between zooxanthellae and coral host leading to enhanced photosynthetic efficiency under high CO$_2$ and nutrient availability (*D'Elia & Wiebe, 1990*; *Muscatine, 1990*). Estimated daily $P_g$ per m$^2$ seafloor for the investigated coral genera, (441–610 mmol O$_2$ m$^{-2}$ seafloor d$^{-1}$), is within the range of other Caribbean corals (67–850 mmol O$_2$ m$^{-2}$ seafloor d$^{-1}$, Fig. 2, *Kanwisher & Wainwright, 1967*), and O$_2$ fluxes of all investigated organism groups are comparable to values reported in the literature (Fig. 2).

Significant spatial differences during non-upwelling were found for CCA with higher productivity at EXP compared to SHE (Fig. 3). These differences may be attributed to the prevailing water current regime in the bay together with high water temperatures during non-upwelling (Tables 1 and 2). An increase in water temperature typically intensifies metabolic activity in CCA (*Hatcher, 1990*; *Littler & Doty, 1975*). However, the lower water flow at SHE (*Bayraktarov, Bastidas-Salamanca & Wild, 2014*) may have prevented the required gas exchange and nutrient uptake, resulting in lower individual CCA productivity at this site. In contrast, the higher rates in individual productivity of algal turfs and CCA at SHE during upwelling (Fig. 3) are potentially a result of the differences in species composition (sensu *Littler, 1973*; *Chisholm, 2003*; *Copertino, Cheshire & Kildea, 2009*; *Ferrari et al., 2012*).

Temporal differences in individual O$_2$ production within the investigated organism groups generally showed two contrary patterns: whereas scleractinian corals on both sites and CCA at EXP produced less O$_2$ during upwelling, algal turfs and CCA at SHE produced more O$_2$. The decreased productivity rates of corals and CCA at EXP during upwelling indicate that low water temperature has an adverse effect on the productivity of these groups. This argument is supported by studies showing that low water temperatures lead to a decrease in photosynthetic performance of primary producers in coral reefs (*Hatcher, 1990*; *Kinsey, 1985*). In contrast, the two-fold higher photosynthetic performance of algal turfs during upwelling may be due to higher nutrient concentrations together with higher water currents during this season (*Bayraktarov, Bastidas-Salamanca & Wild, 2014*; *Bayraktarov, Pizarro & Wild, 2014*), facilitating gas exchange and nutrient uptake. Our findings are supported by *Carpenter & Williams (2007)*, showing that photosynthesis of algal turfs in coral reefs is mainly limited by nutrient uptake, which in turn depends on nutrient availability and water current speed. Whereas productivity of CCA at EXP seems to be temperature-limited, our findings indicate that their productivity at SHE is limited

by nutrient availability as previously suggested for benthic algal communities in water current-sheltered coral reef locations (*Carpenter & Williams, 2007*; *Hatcher, 1990*).

## Contribution of organism-induced $O_2$ fluxes to total benthic $O_2$ production

Our results indicate that the spatial differences in contribution to total benthic $O_2$ production for scleractinian corals, macroalgae, CCA, and microphytobenthos are directly linked to spatial differences in their benthic coverage. For instance, the major contribution of corals (Fig. 4) can be explained by their comparably high benthic coverage (ranging from 24 to 39%; Fig. 2) and highest quantified individual $O_2$ production rates among all investigated groups (Fig. 3). This finding is supported by the estimates of *Wanders (1976a)*, showing that corals accounted for about two-thirds of the total benthic primary production in a Southern Caribbean fringing reef.

Although individual macroalgal production rates were rather low as compared to coral productivity (Fig. 3), the extremely high cover of macroalgae at SHE during upwelling (47 ± 3%, Fig. 2) resulted in macroalgae being the main contributors to total benthic production. Macroalgal cover (incl. the dominant genus *Dictyota*) has previously been found to be particularly high during upwelling (*Bula-Meyer, 1990*; *Cronin & Hay, 1996*; *Diaz-Pulido & Garzón-Ferreira, 2002*) probably due to elevated nutrient concentrations and low water temperatures (*Bayraktarov, Pizarro & Wild, 2014*).

The elevated contributions of corals and CCA at EXP as well as macroalgae and microphytobenthos at SHE during upwelling (Fig. 4) might be due to site-specific differences in abundances (Fig. 2), which in turn are likely caused by site-specific differences in water current regimes (*Bayraktarov, Bastidas-Salamanca & Wild, 2014*; *Werding & Sánchez, 1989*).

Corals, macroalgae, algal turfs, and CCA also exhibited distinct temporal differences in contribution to total benthic productivity. At SHE, corals contributed more to the benthic $O_2$ production during non-upwelling and macroalgae and algal turfs during upwelling, whereas contribution of CCA at EXP was higher during non-upwelling. These differences can be explained with seasonal growth patterns, temperature-dependent changes in individual $O_2$ productivity and temporal shifts in abundances (Figs. 2 and 3). Opposite abundance patterns of CCA and macroalgae are, for example, in agreement with previous studies showing that macroalgae can shade CCA, usually leading to negative correlated abundances of these groups (*Belliveau & Paul, 2002*; *Lirman & Biber, 2000*).

## Total benthic $O_2$ fluxes and ecological perspective

The estimated total daily benthic $O_2$ production at both sites during non-upwelling and upwelling (Fig. 5) are, although comparable, on average slightly lower than the values previously reported for other fore reefs communities (Table 3). These differences might be due to a methodological bias. Whereas previous studies utilized flow respirometry techniques, the current study used incubation methodology, which accounts for production values in the target groups only.

**Table 3** **Mean benthic oxygen production of reef communities and their dominant functional groups of primary producers.** If necessary, original units were converted to $O_2$ estimates assuming a $C:O_2$ metabolic quotient equal to one according to *Gattuso et al. (1996)* and *Carpenter & Williams (2007)*.

| | Location | $P_n$ | $P_g$ | Reference |
|---|---|---|---|---|
| | | (mmol $O_2$ m$^{-2}$ seafloor d$^{-1}$) | | |
| **Reef slope/fore reef communities** | Caribbean | 103–169 | 250–305 | This study |
| | Caribbean | 125–272 | 250–483 | *Eidens et al. (2012)* |
| | Various Atlantic/Pacific | −83–425 | 167–583 | *Hatcher (1988)* |
| | Caribbean | 113–469 | 313–638 | *Adey & Steneck (1985)* |
| **Functional group** | | | | |
| Corals | Caribbean | 227–344 | 441–610 | This study |
| | Caribbean | 328–369 | 441–598 | *Eidens et al. (2012)* |
| | Caribbean | 166 | 447 | *Wanders (1976a)* |
| | Caribbean | | 225–850 | *Kanwisher & Wainwright (1967)* |
| Macroalgae | Caribbean | 117–244 | 198–375 | This study |
| | Caribbean | 244–444 | 375–624 | *Eidens et al. (2012)* |
| | Caribbean | 142–433 | 250–633 | *Wanders (1976b)* |
| | Various Atlantic/Pacific | | 192–3283 | *Hatcher (1988)* |
| Algal turfs | Caribbean | 39–157 | 84–253 | This study |
| | Caribbean | 39–339 | 84–554 | *Eidens et al. (2012)* |
| | Various Atlantic/Pacific | | 75–1008 | *Hatcher (1988)* |
| | Various Atlantic/Pacific | | 83–967 | *Kinsey (1985)* |
| | Caribbean | 175–433 | 308–617 | *Wanders (1976a)* |
| Crustose coralline algae | Caribbean | 44–104 | 58–140 | This study |
| | Caribbean | 44–104 | 58–140 | *Eidens et al. (2012)* |
| | Various Atlantic/Pacific | | 67–83 | *Kinsey (1985)* |
| | Caribbean | 58–117 | 192–258 | *Wanders (1976a)* |
| | Great Barrier Reef | 50–333 | 75–416 | *Chisholm (2003)* |
| Microphytobenthos | Caribbean | 1–67 | 75–143 | This study |
| | Caribbean | 6–87 | 78–191 | *Eidens et al. (2012)* |
| | SW Pacific | 0–8 | 92–150 | *Boucher et al. (1998)* |
| | Various Atlantic/Pacific | | 50–225 | *Kinsey (1985)* |

**Notes.**

Pn, net $O_2$ production; Pg, gross $O_2$ production.

Despite the high spatial and temporal differences in benthic coverage and group-specific $O_2$ fluxes of investigated benthic primary producers as well as their contribution to total benthic productivity, no spatial differences in total benthic $O_2$ fluxes were detected between EXP and SHE. These results were consistent during both non-upwelling and upwelling (Fig. 5). Our findings are supported by *Hatcher (1990)*, showing that the relative coverage of benthic photoautotrophs in a reef community may have little effect on its areal production rate. In TNNP, seasonal differences were only present for $P_g$ at EXP with higher rates during upwelling compared to non-upwelling. These differences are mainly related to individual productivity of algal turfs, being generally two-fold higher during upwelling compared to non-upwelling (Fig. 3), and to the absence of macroalgae at EXP during non-upwelling (Fig. 2). This is in agreement with studies by *Kinsey (1985)* and

*Hatcher (1990)*, reporting that algae, as one of the most seasonal component in coral reefs, account for seasonal shifts in benthic reef productivity.

The lack of seasonality of $P_n$ and $P_g$ regarding communities at SHE as well as $P_g$ at EXP stands in contrast to earlier studies (*Eidens et al., 2012*; *Kinsey, 1977*; *Kinsey, 1985*; *Smith, 1981*), which found an approximately two-fold difference in benthic primary production between seasons. This lack of seasonality in $P_n$ and partly in $P_g$ might be related to seasonal changes of abiotic factors in TNNP that compensate for each other (Table 1). The observed similarity in productivity rates during different seasons suggest that coral reefs in TNNP can cope with pronounced seasonal variations in light availability, water temperature, and nutrient availability. Nevertheless, total $P_n$ and $P_g$ during the upwelling in 2010/2011 ($P_n$: 244–272 and $P_g$: 476–483 mmol $O_2$ m$^{-2}$ seafloor d$^{-1}$) were not only higher compared to non-upwelling (*Eidens et al., 2012*) but also higher than during the subsequent upwelling in 2011/2012 (Fig. 5). These findings suggest that interannual variations affect the productivity of TNNP coral reefs. Dramatic ENSO-related water temperature increases and high precipitation in the study area (*Bayraktarov et al., 2013*; *Hoyos et al., 2013*) led to coral bleaching at the end of 2010 (*Bayraktarov et al., 2013*). Surprisingly, bleached corals in the bay recovered quickly in the course of the following upwelling event (*Bayraktarov et al., 2013*) and exhibited similar $O_2$ production rates during all study periods (*Eidens et al., 2012*), indicating a high resilience of TNNP corals. Moreover, macroalgae and algal turf seemed to benefit from the environmental conditions during the upwelling following the ENSO-related disturbance events, resulting in higher group-specific productivity during the upwelling in 2010/2011 compared to subsequent study periods (*Eidens et al., 2012*). The elevated production rates of macroalgae and algal turfs together with the quick recovery of corals from bleaching likely accounted for a higher benthic productivity during the upwelling in 2011/2011 compared to non-upwelling (*Eidens et al., 2012*) and the upwelling in 2011/2012 (Fig. 5). These findings indicate that extreme ENSO-related disturbances do not have long-lasting effects on the functioning of local benthic communities in TNNP.

In conclusion, the present study showed that total benthic productivity in TNNP is relatively constant despite high variations in key environmental parameters. This stable benthic productivity suggests a relatively high resilience of local benthic communities against natural environmental fluctuations and anthropogenic disturbances. We therefore recommend that TNNP should be considered as a conservation priority area.

## ACKNOWLEDGEMENTS

We thank JF Lazarus-Agudelo, JC Vega-Sequeda, T Deuß, M Kabella, R Kügler, and J Rau for assistance during fieldwork and the staff of Instituto de Investigaciones Marinas y Costeras (INVEMAR) for logistic support. We acknowledge the kind collaboration of the administration and staff of the Tayrona National Natural Park. Furthermore, the authors would like to thank Cajo ter Braak and four anonymous reviewers for their useful comments that improved the manuscript significantly.

### Funding

This research was funded and supported by the German Academic Exchange Service (DAAD; http://www.daad.de/en/) through the German-Colombian Center of Excellence in Marine Sciences (CEMarin; http://www.cemarin.org/) and by the German Research Foundation (DFG; http://www.dfg.de/en/index.jsp) grant Wi 2677/6-1 to C Wild. The funders had no role in study design, data collection and analysis, decision to publish, or preparation of the manuscript.

### Grant Disclosures

The following grant information was disclosed by the authors:
German Academic Exchange Service.
German-Colombian Center of Excellence in Marine Sciences.
German Research Foundation: Wi 2677/6-1.

### Competing Interests

The authors declare there are no competing interests.

### Author Contributions

- Corvin Eidens conceived and designed the experiments, performed the experiments, analyzed the data, wrote the paper, prepared figures and/or tables, reviewed drafts of the paper.
- Elisa Bayraktarov conceived and designed the experiments, performed the experiments, wrote the paper, reviewed drafts of the paper.
- Torsten Hauffe analyzed the data, wrote the paper, prepared figures and/or tables, reviewed drafts of the paper.
- Valeria Pizarro performed the experiments, wrote the paper, reviewed drafts of the paper.
- Thomas Wilke contributed reagents/materials/analysis tools, wrote the paper, reviewed drafts of the paper.
- Christian Wild conceived and designed the experiments, contributed reagents/materials/analysis tools, wrote the paper, reviewed drafts of the paper.

### Field Study Permissions

The following information was supplied relating to field study approvals (i.e., approving body and any reference numbers):

All necessary permits (DGI-SCI-BEM-00488) were obtained by Instituto de Investigaciones Marinas y Costeras (INVEMAR) in Santa Marta, Colombia which complied with all relevant regulations.

## Supplemental Information

Supplemental information for this article can be found online at http://dx.doi.org/10.7717/peerj.554#supplemental-information.

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
