# Peer review of "Benthic primary production in an upwelling-influenced coral reef, Colombian Caribbean"

_PeerJ, doi:10.7717/peerj.554_

## Round 0.1 · original submission · Major Revisions

· Academic Editor

Major Revisions

The title is too grand, spatial and temporal variation suggests more than 2 sites and 2 times. What about “benthic primary production in an upwelling-influenced Colombian Caribbean coral reef”.

You should scale down the conclusions; one reviewer said “conclusions are too sweeping”. One example is the last sentence of the abstract: Do the findings indicate adaptation? No, you interpret it as such.

From the statistical side, it is unclear what the status is of the replicates in the supplemental xlsx file. They come from the sites/times as described (4 units), then different taxa and then different replicates of the same taxon. How are taxa and replicates phyisically done? In different incubation containers. Does it represent more than just technical replication?

The Supplemental table do not have a caption/description. Was the ANOVA wrt to a factor with 4 levels or wrt to two factors (site and time) with an interaction. It should be the interaction that is of interest. The site differ before upwelling and there are logical difference in time, so the upwelling effect is an interaction. See the literature on the before after control impact (BACI) literature(e.g. Underwood 1992).


Spatial differences -> difference between the


Underwood AJ (1992) Beyond BACI: the detection of environmental impacts on populations in the real, but variable, world. JExp Mar Biol Ecol 161:145-178

Reviewer 1 ·

Basic reporting

No significant comments...conclusions are too sweeping, but not worth arguing over.

Experimental design

No problems.

Validity of the findings

No problems.

Reviewer 2 ·

Basic reporting

This is an interesting study and one that I think should be publishable in PeerJ after major revision. The authors were interested in quantifying the spatial and temporal variability in benthic production on coral reef communities before and after a large upwelling event. While the paper is generally well written I think the experimental design needs to be more clearly articulated and the authors should clearly identify their hypotheses.

Experimental design

The authors used a combination of individual species incubations in the lab with functional group abundances in the field and 2D-3D relationships to scale up experimental results to larger reef scale processes. While I like this idea in principle I am not convinced that it works in practice without having some actual in situ data collected. Did the authors measured production and respiration in situ? Were there differences across space and over time that may be attributable to the differences in species assemblages? Were there any organisms present that were not accounted for in the incubation experiments? Cryptic fauna, etc.?

I would like to see more data presented on the individual organism incubation results. The means and standard errors should be plotted for Pnet and Pgross for all the groups across space and over time in a figure.

It is not clear how the incubation data were scaled up to the reef scape. For macroalgae how would you account for tissue that is overlapping or hidden under corals or other taxa? Biomass seems like it would have been a better metric to normalize your estimates to (you can harvest seaweed in plots with differing percent cover values to generate a relationship between percent cover and biomass and then repeat this between seasons).

Presumably the expectations were that production rates should be higher under upwelling conditions associated with higher inorganic nutrient availability. Why weren't nutrients measured to validate differences seen among taxa between sampling times?

Validity of the findings

I think the authors should place more emphasis on the findings from the incubation experiments and less on the reef scale patterns as this seems to be a bit speculative. The authors need to clarify how these reef scape results were generated and could be repeated.

Additional comments

Please clarify the goals and hypotheses as well as some of the more detailed methods.

Reviewer 3 ·

Basic reporting

The authors measured oxygen flux from an array of benthic guilds (corals, turf algae, macroalgae, crustose coralline algae, and microphytobenthos) in incubation chambers. To do this, they collected the organisms from two areas of a bay in Colombia, one nominally more and one less exposed to wave activity and upwelling, although no data to support this are shown. Organisms were either broken and epoxied onto plates and set out to recover (coral) or harvested just prior to incubation (other guilds) and oxygen evolution measured in chambers that were manipulated to ensure similarity to ambient light and temperature levels. The study was conducted in November and March of subsequent years, straddling an upwelling event that is not characterized herein, but is defined by the authors as the driver of trends observed in the oxygen flux data. Benthic community composition was also characterized by point transects at both time points and locations and used to generate areal oxygen fluxes in combination with the incubation fluxes.
Major findings include significant differences in oxygen flux in space with crustose coralline algae, and in time with corals, macroalgae, and CCA, but the latter changes were not similar in direction. Corals contributed more areal oxygen flux than the other guilds.
The statistical outcomes of the study are in the supplemental material, making it hard to interpret the results. Including indications of significance in the Tables of results they have included in the work (eg the conventional a, b, c superscripts after values) would be beneficial.
Editing the discussion and refocusing on data contained in the paper (oxygen fluxes in chambers and areal) would make the paper stronger, and may yield an interesting story in and of itself while avoiding the issue of lacking environmental data.

Experimental design

The main aim of the paper is to link seasonal environmental variation to biotic responses and change. However, as the environmental characterization data the paper refers to was collected in one one-month period in late 2011 and one one-week period in early 2012 and carried out at two sites within one bay, there is insufficient data to conclude what is driven by season/upwelling and what is natural variation due to other factors. The authors could avoid these issues by expanding on the experiment at greater spatial and temporal scales.
The experiment is also founded upon incubations of a variety of organisms in tanks. While the authors went to great lengths to avoid bottle effects in this, the experiment suffers from a large degree of handling of each of the organisms and that most organisms were handled differently to each other. As no controls for this handling were provided in the paper it is unclear what results were due to artifacts and what were due to natural variation. Conducting or providing the appropriate controls would reinforce the experimental design markedly.
Further, the data gathered in the study are markedly unbalanced (n~22 vs n~3), and it is unclear from the paper how this was handled. Indicating how unbalanced data was accommodated, or repeating the statistical analysis in a manner that accommodates unbalanced data would remedy this and make the conclusions more sound.

Validity of the findings

Due to the limited nature of the study, the authors were not able to provide logically supported findings from their work. If the experiment were expanded spatially and temporally, and the new environmental data presented with the new oxygen flux data set then it is promising that compelling and supported findings would be forthcoming.

Additional comments

Minor Comments:
Title The “…on Colombian Caribbean Coral Reefs” should read “…on a Colombian Caribbean Coral Reef.”
41 What does “where diversely structured coral communities are present” mean?
44 “The area thereby provides…” would be easier to read.
45-46 “…key coral reef ecosystem service productivity…” is awkwardly worded.
87 “from NE to SW” would help.
196 Were interactions allowed in the 2-Way ANOVA? How was unbalanced data accommodated?
233 “…season, the contribution…”
239 “The corals’ contribution…”
Discussion Frequent qualified sentences. eg 303-304 “…EXP could further have negatively affected coral productivity as decomposition of macroalgae may result in toxicity…”
392 & 394 “On the one hand”/”On the other hand” colloquial language.
396 “…thus promoting photosynthesis…”
Table 1 Formatting makes it hard to read.
Table 2 appears to have two missing values.

---

## Round 0.2 · Minor Revisions

· Academic Editor

Minor Revisions

The statistical analyis is now state-of-the-art. I did not understand the unbalancedness motivation on line 202. So start the sentence just with: We used...

In the revision please take care of the Basic reporting issues noted by Reviewer 4 and try answer the claim "Weakness of the discussion is that predominantly lit references are invoked which comply best with the obtained data.". So are there references that comply less? If so, that should be reported and is, of course, no issue for acceptance. Consider mildly shortening the discussion.

Minor points:

The title above the abstract is still the old one.

L 184: corrected with? How? The rest of the sentence talks about subtraction which is clear, but why then the unspecified "corrected"?

Reviewer 4 ·

Basic reporting

The MS is overall reasonably well written. It is however not always clear what the authors try to say.
For instance see line 432… interannual influences??? Meaning variations??
437… similar specific O2 etc compared to subsequent????
444.. upwelling in 2011/2011??

Experimental design

The choice to incubate in small incubators (different sizes) for short periods of time is definitely not ideal. The extrapolation of fluxes of functional groups to cover area and arrive at benthic community metabolisms is not really plausible. All these aspects influences the outcome of the experiments.

Validity of the findings

Authors made a series of observations on O2 production of different functional groups of benthic organisms before and after upwelling. Furthermore they extrapolated the results to the reef surface area taking account of the cover of different functional groups. They found significant differences in O2 production in time and space for several functional groups. Upwelling significantly enhanced the net production of algal turf irrespective of location. Net production of corals dropped in the upwelling but Pg was constant. Upwelling did not affect the productivity of macroalgae and microphytobenthos. Macroalgal O2 production increased per m-2 reef because algal cover increased.
Problem is that the differences between functional groups basically cannot be explained further than that “season” and “site” apparently effects the O2 fluxes. Many variables were different between the tested periods e.g. time per se (Nov 2011- March 2012= >100 days), light availability, temperature, water movement, nutrient availability, turbidity etc. . Moreover the CCA and algal turf community composition may have been different between different incubation experiments. Therefore I have concerns regarding the extrapolation of findings to the field situation.
Weakness of the discussion is that predominantly lit references are invoked which comply best with the obtained data. This does not really contribute to a better understanding. After all it might well be the complete set of variables (with interactions) which changes over time that affects the O2 productivity. Observations are of interest, but the speculations are not really providing the reader with new insights. What is interesting to note is that the respiration varies with season.

Additional comments

The study is of local interest. Therefor I would suggest to submit it to a local journal. Further I would recommend to seriously cut down speculations from the discussion and shorten the discussion.

---

## Round 0.3 · accepted · Accept

· Academic Editor

Accept

The recommmended changes have been carried through and the paper is ready to go forward.